

# Forest habitats and plant communities strongly predicts Megachilidae bee biodiversity

Lindsie M. McCabe[1,2], Paige Chesshire[2] and Neil S. Cobb[3]

[1] USDA-ARS Pollinating Insects Research Unit, Logan, Utah, United States
[2] Department of Biological Science, Northern Arizona University, Flagstaff, Arizona, United States
[3] Biodiversity Outreach Network, Flagstaff, AZ, United States

## ABSTRACT

Megachilidae is one of the United States' most diverse bee families, with 667 described species in 19 genera. Unlike other bee families, which are primarily ground nesters, most megachilid bees require biotic cavities for nesting (*i.e.*, wood, pithy stems, *etc.*). For this group, the availability of woody-plant species may be as important as nectar/pollen resources in maintaining populations. We studied Megachilidae biodiversity in the continental United States. We confirmed that the highest species richness of Megachilidae was in the southwestern United States. We examined the relationship between species richness and climate, land cover, tree species richness, and flowering plant diversity. When examining environmental predictors across the conterminous United States, we found that forested habitats, but not tree diversity, strongly predicted Megachilidae richness. Additionally, Megachilidae richness was highest in areas with high temperature and low precipitation, however this was not linearly correlated and strongly positively correlated with flowering plant diversity. Our research suggests that the availability of nesting substrate (forested habitats), and not only flowering plants, is particularly important for these cavity-nesting species. Since trees and forested areas are particularly susceptible to climate change, including effects of drought, fire, and infestations, nesting substrates could become a potential limiting resource for Megachilidae populations.

# INTRODUCTION

Insect pollinators provide key ecosystem services; over 86% of all flowering plant species depend on insects for pollination (*Klein et al., 2006*; *Ollerton, Winfree & Tarrant, 2011*), and bees alone pollinate roughly one-third of the world's staple food crops (*Potts et al., 2010*). Reported declines in bee species abundance may affect both economically important crop yield and wild plant populations worldwide (*Brown & Paxton, 2009*; *Potts et al., 2010*; *Sahli & Conner, 2006*). Bee declines can be attributed to habitat loss, pesticides, invasive species, and climate change (*Klein et al., 2007*; *Memmott et al., 2010*; *Potts et al., 2016*). Despite the importance of bees, most bee species distributions are unknown, and even less is known about the factors that drive their distributions.

Corresponding author
Lindsie M. McCabe,
lindsie.mccabe@usda.gov

Bees are highly diverse and have a wide range of life history strategies including sociality, nesting strategies, parasitism and diets (*Michener, 2007*; *Westrich, 2007*; *Danforth et al., 2019*; *Mikát, Matoušková & Straka, 2021*; *Bossert et al., 2022*). Many of these traits have evolved and/or been lost over evolutionary history and geographic space, contributing to the diversification of bees (*Danforth et al., 2013*; *Hedtke, Patiny & Danforth, 2013*). Climate, seasonality and shifting plants have all contributed to species and trait diversification for this group (*Toussaint et al., 2012*; *Kergoat et al., 2018*). One unifying characteristic of bees, however, is that bees have strong biotic interactions, such as relying on pollen and nectar from plants for food (*Bartomeus et al., 2013*). Despite these strong ties, floral richness has not been well associated with bee species richness globally; the tropics are a hot bed of diversity for plant species but have the lowest bee species diversity (*Michener, 2007*; *Orr et al., 2020*). This suggests that other life history traits are strongly influencing geographic distribution for this taxa.

Megachilidae is globally distributed and comprises 35% of the known 4,000 bee species (*Michener, 2007*) in the United States. Megachilid bees have important roles in pollination of a diverse range of ecosystems, including natural, urban, and agricultural environments (*Gonzalez et al., 2012*). *Osmia* and *Megachile* are successfully managed for crop and wildflower pollination; and *Osmia lignaria* is the key pollinator reared for apple, cherries, and other fruit trees *Megachile rotundata* is reared as the primary pollinator of alfalfa (*Bosch & Kemp, 2002*; *Boyle & Pitts-Singer, 2019*; *Pitts-Singer et al., 2018*; *Pitts-Singer & Cane, 2011*; *Sampson & Cane, 2000*).

Many megachilid bees nest in pre-existing cavities in decaying wood, hollow plant stems, or hollow twigs, and use a range of materials to create their nest cells, including masticated leaves, mud, plant resins, or flower petals (*Bosch & Kemp, 2002*; *Litman et al., 2011*; *Young et al., 2016*). The wide use of foreign materials for nest building is a likely reason for evolutionary range expansion and high diversification in Megachilidae, (*Gonzalez et al., 2012*; *Litman et al., 2011*). About 75% of the genera for this family have above ground/cavity-nesting traits (*Eickwort, Matthews & Carpenter, 1981*), however primitive forms of nesting (*i.e.*, ground borrowing) still exist in eight of the 19 genera (*Bosch, Maeta & Rust, 2001*). It is likely that the above ground nesting behavior was repeatedly evolved and lost numerous times throughout the evolution of this group, since very closely related species can have drastically different nesting behaviors (*Sedivy, Dorn & Müller, 2013*).

Bees' habitat, shelter, and food may all be impacted by climate change events (*Corbet et al., 1993*; *Tews et al., 2006*; *Bartomeus et al., 2011*; *McCabe, Aslan & Cobb, 2022*). When assessing how climate change and other anthropogenic disturbances will impact these species, research should not just look at how increasing temperatures will directly affect bees. Understanding where species are distributed can help to understand what the most limiting resources are for the success of a population.

Recent aggregation of databases and occurrence records have made it possible to assess spatial and temporal questions about insect species diversity (*Cavender-Bares et al., 2009*). This data can provide insight on how species and communities will respond to future disturbance (*i.e.*, climate, habitat loss, *etc.*), but it is first critical to understand what is

driving patterns of biodiversity and how these species respond to different environmental parameters (*Weiher et al., 2011*).

While it is widely accepted that Megachilidae have strong ties to biotic associations (*e.g.*, host plants and cavity nesting resources: trees and pithy stems) in North America (*Martins & de Almeida, 1994*; *Michener, 2007*; *Bosch, Maeta & Rust, 2001*; *Cane, Griswold & Parker, 2007*), no study has systematically assessed the relationship between species richness and nesting substrate. We reviewed the biogeographic patterns of United States Megachilidae species, with two primary questions. (1) How can biogeographic patterns be explained by climate, nesting resources, and floral resources? (2) Are there specific relationships between Megachilidae biodiversity hotspots and taxonomic and/or morphological structure of woody plants? We predicted that for Megachilidae in particular they would be distributed in areas where cavity-nesting resources were also abundant, and in areas with high plant diversity.

## METHODS

### Data extraction

Bee data was collected from three aggregate databases: iDigBio, Symbiota Collection of Arthropod Network (SCAN) (*SCAN, 2019*), and Global Biodiversity Information Facility (GBIF) (*Global Biodiversity Information Facility, 2020*). For all three databases, all records from the family "Megachilidae" occurring in the "United States", "Canada" and "Mexico" were extracted. These records were thoroughly cleaned to only include those with georeferenced information, identified to species (if they record was lacking scientific name or classified as a morphospecies they were not included), and were determined to be specimen records; observation records were not used in this data set (records such as iNaturalist observations, BugGuide and/or Xerces society records that appeared on GBIF or SCAN were not used);. All duplicate records were removed by filtering each dataset using the Darwin Core Archive (DwC) field "catalog number" and "institution code". Records were then checked for taxonomic (including synonyms) and geographical errors. We resolved all spelling, male/female suffix discrepancies and all additional information that may have been included in this field. All species were vetted for location accuracy and probability that the species occurred in the area sampled by taxonomic experts of this family (T. Griswold & J. Ascher, 2021, personal communication). Data were cleaned in the same manner as in *Chesshire et al. (2023)* and as in *Dorey et al. (2023)*. Great care was taken to make sure all records were as accurate as possible. There was significant lack of occurrence records for north and central Canada, and patches of minimal records throughout Montana and the Dakotas in the United States, which is likely due to sampling biases, however data was restricted only to the United States due to insufficient land cover data available in Canada and/or Mexico. In total we used 248,367 cleaned georeferenced records that comprised 667 Megachilidae species for the United States and represented 19 genera (Table 1). All Megachilidae species were used in these analyses regardless of nesting preference since nesting preference is so wildly variable between related species and species level nesting preferences were available for less than 20% of all Megachilidae species.

Table 1 List of genera and their known nesting substrates, number of occurrence records used in this analysis, and average number of plants visited based on known literature and DiscoverLife.

| Genus | # of species | Number of occurrence records |
| --- | --- | --- |
| Anthidiellum | 5 | 2,660 |
| Anthidium | 33 | 13,278 |
| Ashmeadiella | 51 | 23,752 |
| Atoposmia | 23 | 1,837 |
| Chelostoma | 10 | 3,604 |
| Coelioxys | 37 | 7,601 |
| Dianthidium | 22 | 8,664 |
| Dioxys | 4 | 383 |
| Heriades | 11 | 9,325 |
| Hoplitis | 55 | 16,111 |
| Lithurgopsis | 6 | 1,519 |
| Lithurgus | 1 | 288 |
| Megachile | 116 | 66,676 |
| Osmia | 129 | 83,445 |
| Paranthidium | 1 | 783 |
| Protosmia | 1 | 2,965 |
| Pseudoanthidium | 1 | 55 |
| Stelis | 40 | 3,333 |
| Trachusa | 15 | 2,088 |

Climatic variables were extracted from WorldClim (*Fick & Hijmans, 2017*). All 19 climatic variables were downloaded and used in our initial analysis. Vegetation and Canopy cover were extracted from LandFire (*Rollins, 2009*). The LF EVT 14 (LandFire Existing Vegetation Type 2014) dataset was used to distinguish vegetation habitat classification effects on richness. Locality of the bee occurrence records determined the extraction point information from LandFire and WorldClim. All data was upscaled to a $60 \times 60$ km ($12,100$ km$^2$) resolution; For WorldClim data we took the average among cells and for LandFire data we took the majority values for each cell to scale up to a courser resolution, using methods developed by *Gann (2019)* LandFire land cover classifications were used in two resolutions; firstly, we used the EVT_LF group, which splits landcover into eight distinct land cover categories, such as tree, agriculture, shrubland. Secondly, for those points that occurred in the "tree" land cover type, we used the EVT_PHY land cover category, which has finer land cover classifications. This group included 31 total land cover types, seven of which occurred in the tree cover type (Supplemental 1). Flowering plant richness was roughly assigned to regions based on *Barthlott et al. (2007)* data, and since no large-scale data for the United States exists, methods for assigning flower diversity to regions were based on *Bystriakova et al. (2018)*. Additionally, tree diversity was derived from *Wilson, Lister & Riemann (2012)* data sets and upscaled to the same resolutions as the other factors.

## Richness map

Occurrence data records were aggregated to a 60 × 60 km resolution raster. Duplicate species data points that fell into the same 60 × 60 km squares were aggregated to one point (keeping just richness values). Rasters were generated by rasterizing the occurrence records using the raster package in R (*Hijmans & van Etten, 2014*). This meant that for each grid cell, an occupied value was only included if one or more occurrence records were found within that cell. Rasters were then summed together vis RasterStack to create a map of overall observed richness for the entire Megachilidae family. Due to sampling biases around roads, developed areas and areas that are home to large museum collections (*Cobb et al., 2019*), a 60 × 60 km spatial resolution was selected. Finer resolutions increased the number of grid cells that has zero points and coarser resolutions compromised the habitat preferences. Even while using a coarse resolution, roughly half of the grid cells had no species recorded. However, despite this resolution selection there were still a number of pixels that have less than 10 species. Any pixel that had less than ten species (presumably due to sampling bias) were removed from the data set. Unfortunately, do to sampling incompleteness, many of these zero values occurred in the Midwest and Nevada.

To account for low sampling in these areas, we broke up the United States into regions, and samples the North East, Southern East, Midwest, Northwest, and Southwest to analyze richness and landcover effects only.

## Statistical analyses

To test the relationship between climate variables and richness we ran a generalized linear model with a negative binomial distribution (GLMnb) that included all eight of the selected climate variables. Prior to this analysis we ran a correlation test to assess the relationships between the 19 WorldClim variables (Supplemental 2). If one of the bioclimatic variables was highly correlated with another one, we selected the variable we believed would have the greatest ecological impact on these taxa. Additionally, we ran another GLMnb to test the relationship between vegetation cover and Megachilidae richness. We then ran a least squares mean *post hoc* test to determine the relationship between each of the vegetation types. Regional-level centroid averages should not show autocorrelation (verified using Morans I) the models above could be used for assessing variable importance and species richness. In addition to the GLMnb test that was performed on the entire United States, we ran an additional analysis where both landcover (vegetation type) and region were compared to species richness. We then ran a least squares mean *post hoc* test to determine the relationship between each of the vegetation types for each region. We did not compare regions to one another, only vegetation type within a region. We performed a non-metric multidimensional scaling ordination (NMDS) using a Bray-Curtis distance matrix to visualize differences in species among land cover types. All analyses were done in R 4.0.3 (*R Core Team, 2013*) using packages MASS for GLMnb (*Ripley et al., 2013*), package Corrplot for correlation analysis (*Wei et al., 2017*) and Vegan for NMDS (*Dixon, 2003*).

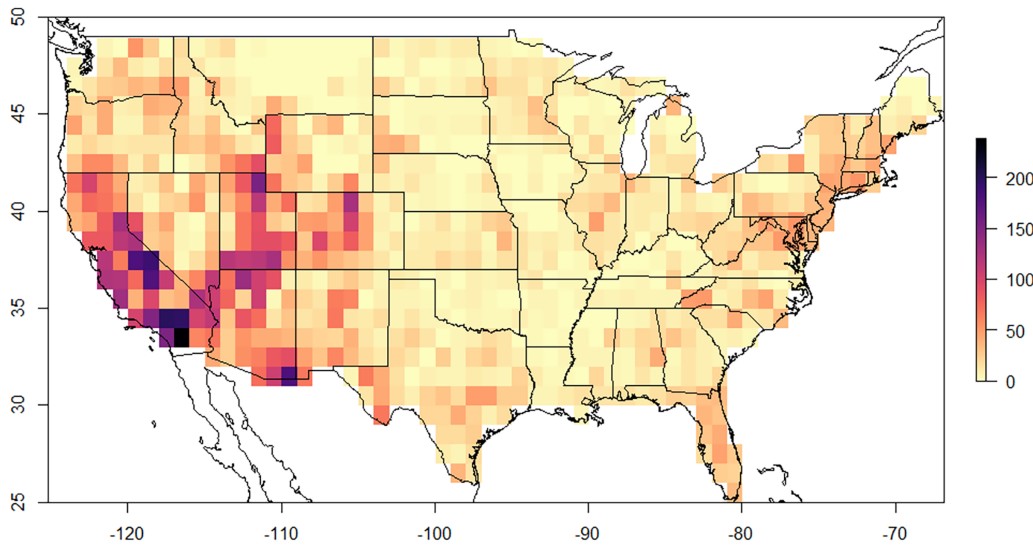

**Figure 1** **60 × 60 km resolution (3,600 km²) of Megachilidae species diversity across the lower 48 United States.** Areas with high diversity are indicated by purple to black, while those areas with low diversity are indicated by light yellow or orange colors.

## RESULTS

There was one hotspot of biodiversity identified in the United States for Megachilidae bees. The southwestern United States by far had the greatest species richness over any other area in the US (Fig. 1). Average richness in the southwest was 43.5 +/− 26.12 species, while in the southeastern US it was 19.03 +/− 8.64 species, northeast was 19.36 +/− 8.12, Midwest was 16.81 +/− 7.1 species and northwest was 21.59 +/− 12.43 species per 60 × 60 km area. The greatest species richness was in southern California near Los Angeles with 253 species for the 60 × 60 km area. Higher elevation mountain regions, including the Sierra Nevada Mountain Range, the Colorado Plateau and the Wasatch Mountains were the most diverse areas within the western region and overall United States.

Richness was predicted by three factors: land cover (*i.e.*, tee cover), climate, and flowering plants (Table 2). Land cover type was the most correlated environmental factor in explaining bee richness (f = 28.57 $p < 0.001$). Areas that were classified by forested habitats (trees) had twice as much diversity than any other habitat with 44.75 +/− 25.53 species per given pixel (Fig. 2). Shrub and herb dominated habitats had the second highest number of species rich pixels with roughly 25.03 +/− 17.56 and 27.10 +/− 23.48 average species per pixel, respectively. Herb and shrub habitats were not significantly different from one another ($p = 0.185$). Agriculture, developed, barren and sparse habitats had averages between 10–15 species per pixel; none of these land cover types were significantly difference from one another (Table S2). Despite land cover type being highly correlated with Megachilidae richness, overall tree diversity did not have a significant effect in predicting Megachilidae richness (z = 1.321, $p = 0.187$).

The southwestern United States drove the pattern for having the most species in treed/forested habitats. In the midwest, the northwest and southeast none of the landcover types were different from one another (Table S2, Fig. 2). All habitats had an equal number

**Table 2 Key environmental predictors of Megachilidae richness and general linear models fitted to the species richness of Megachilidae to the United States extent.**

| Variable | Acronym | Units | Source | z-value | p-value |
|---|---|---|---|---|---|
| Annual mean temperature | BIO1 | °C | BIOCLIM (*Hijmans et al., 2005*) | 4.559 | <0.001 |
| Mean diurnal range | BIO2 | °C | BIOCLIM (*Hijmans et al., 2005*) | 10.45 | <0.001 |
| Max temperature of the warmest month | BIO5 | °C | BIOCLIM (*Hijmans et al., 2005*) | 0.883 | 0.377 |
| Mean temperature of the wettest month | BIO8 | °C | BIOCLIM (*Hijmans et al., 2005*) | −11.13 | <0.001 |
| Mean temperature of the driest quarter | BIO9 | °C | BIOCLIM (*Hijmans et al., 2005*) | 23.15 | <0.001 |
| Annual mean precipitation | BIO12 | mm | BIOCLIM (*Hijmans et al., 2005*) | −12.57 | <0.001 |
| Precipitation seasonality | BIO15 | mm | BIOCLIM (*Hijmans et al., 2005*) | 8.774 | <0.001 |
| Precipitation of the warmest quarter | BIO18 | mm | BIOCLIM (*Hijmans et al., 2005*) | −16.65 | <0.001 |
| LandFire existing vegetation type | EVT_LF | Categorical classification | LandFire 2016 | 28.57 | <0.001 |
| LandFire existing vegetation type subclass forest types | EVT_SBCLS | Categorical classification | LandFire 2016 | 13.68 | <0.001 |
| United States tree diversity | TreeDiv | Number of species per grid cell | adapted from *Wilson, Lister & Riemann (2012)* | 1.321 | 0.187 |
| United States herbaceous flowering plant diversity | PlantDiv | Number of species per grid cell | adapted from *Barthlott et al. (2007)* | 4.742 | <0.001 |

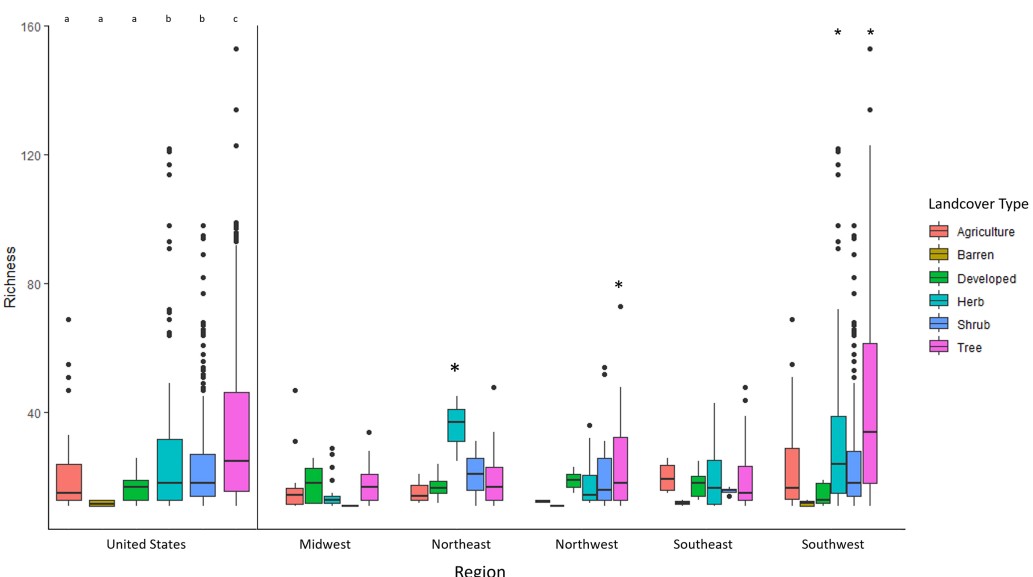

**Figure 2 The average Megachilidae richness for each LandFire classification type (EVT_LF).** Letters and/or asterisk denote significance.

of species per pixel. However, the southwestern United States had at least double the species in treed habitats than it did in agriculture, barren, developed, and shrub habitat (F = 30.702, $p < 0.001$, Fig. 2). Additionally, shrub and herbaceous habitats had more richness than agriculture, barren and developed lands.

All eight climatic variables selected showed a significant effect in predicting richness except for maximum temperature of the warmest quarter ($z = 0.883$, $p = 0.337$). Mean temperature of the driest quarter explained 4x as much variance than any other climatic variable ($z = 23.15$, $p < 0.001$). While mean temperature of the warmest quarter and precipitation seasonality were significant in the model, they explained less than 1% of the variance combined ($z = -11.33$, $p < 0.001$ & $z = 8.774$, $p < 0.001$ respectively). Herbaceous flowering plant diversity strongly predicted Megachilidae richness; as flowering plant diversity increased, so did Megachilidae diversity ($z = 4.742$, $p < 0.001$). Flowering plant richness also peaked in the southwestern United States. We did not examine specific associations for this analysis.

## DISCUSSION

Our results indicate that Megachilidae biodiversity has its hotspot in the southwestern United States, this is consistent with geographic patterns found by *Michener (2007)*. Additionally, this is true for many bee species outside of Megachilidae, as well, since the southwestern United States is also a hotspot for bee biodiversity on a global scale (*Griswold et al., 2018*; *Orr et al., 2020*). The southwest is typically associated with dry air climates that are more suitable for ground nesting bees and thus hypothesized to harbor the greatest bee diversity globally, with around 80% of those being ground nesters (*Danforth et al., 2019*). However for Megachilidae, where 70% of the genera are cavity nesters, the same environmental traits may not be as crucial (*Cane, Griswold & Parker, 2007*). Importantly, the majority of cavity nesting species in tribe Osmiini are concentrated in North America (*Praz et al., 2008*), while many highly related *Megachile* species in the Palearctic still nest in the ground or have diversified nesting strategies (*Danforth et al., 2004*; *Patiny, Michez & Danforth, 2008*). It is hypothesized that cavity nesters were better able to survive trips across long water barriers, potentially explaining why it is mostly these Osmiini species inhabiting the new world (*Michener, 2007*). Therefore, some of the climate preferences and/or floral dependencies may be carried over from this lineage of soil nesting bees.

What makes United States Megachilidae so unique, however, is their strong ties to biotic nesting resources. Contrary to what *Orr et al. (2020)* found for all bee species, here we show that Megachilidae biodiversity was highly correlated with forested habitat, particularly in the southwestern United States, suggesting that these areas of more heterogeneous landscapes could be driving species diversification for this group (*Hedtke, Patiny & Danforth, 2013*). Groups such as Colletidae have shown stronger correlations with floral resources and climate than they have nesting requirements (*Bystriakova et al., 2018*). Other groups like *Bombus* also seem to be more strongly tied to floral resources for driving biogeographical patterns (*Williams, Lonsdorf & Ward, 2014*). Our study is not the first to show that bees may be restricted in distribution due to nesting resources or habitats (*Gordon, 2000*; *Romero, Ornosa & Vargas, 2020*). Additionally, traditional range models only factor in climate analyses however it is likely that landscape/ habitat play a large role in determining species geographic ranges (*Graham & MacLean, 2018*; *Palma et al., 2016*). *McCabe et al. (2020)* found that, along an elevation gradient in the southwest US Megachilidae were distributed in higher cooler elevations. These higher elevation

environments also correspond with habitats that are forested. This pattern has only been documented in one other bee group, *Bombus* (*Williams et al., 2014*). However, *Bombus* is a social ground nesting species and has much lower diversity than the Megachilidae family.

Megachilidae showed a strong preference to floral diversity, as would be expected. Increased plant richness and diversity in an area, which likely also equates to increased floral abundance, can lead to greater floral resource availability (*Blaauw & Isaacs, 2014*; *Crone & Williams, 2016*). However, it was impossible to link specific flora types with drivers of Megachilidae biodiversity since many host plant associations are unknown. Even where host plant specialization has been well studied, it is unclear how the distribution of those plants plays a role in the bee species distributions (*Michener, 2007*). Although there are limitations for plant-insect interactions, open forest habitat with understory vegetation or open meadows with floral resources promotes bee diversity (*Mola et al., 2021*). Likewise, areas of increasing canopy cover in the southwest have been shown to decrease bee species richness, but Megachilidae species richness increases (*McCabe et al., 2019*).

Nationwide, climate analysis showed a trend towards warmer and drier environments. Megachilidae richness increased with increasing mean annual temperatures and decreasing mean annual precipitation, which is consistent with trends found in other bee species (*Michener, 2007*; *Orr et al., 2020*). However, analyses with the more nuanced climate variables revealed even more informative patterns for Megachilid species. Mean temperature of the wettest quarter was important in predicting richness; as mean temperatures decreased, Megachilidae richness increased. This is likely because species in the Megachilidae family often require a minimum number of cold days in order to complete diapause (*Bosch, Sgolastra & Kemp, 2010*) and the number of cool days during overwintering can directly influence their emergence timing (*Sgolastra & Kemp, 2010*). Unfortunately, future warming will likely change the distribution patterns of many Megachilidae species. *McCabe, Aslan & Cobb (2022)* found that for a community of cavity nesting species in Arizona, warming overwintering temperatures negatively affected emergence rates. Although the species were able withstand cooler temperatures in higher elevations, there were no nesting resources available in those environments.

Megachilidae may have more restrictions when adapting to climate change due to their associations with cavity nest availability (*e.g.*, dead and down wood, pithy stems, *etc.*) and floral resources availability (*Cane, Griswold & Parker, 2007*). This dependency on nesting cavities adds an additional level of complexity when asking the question of how they will respond to climate change. In fact, it is estimated that 98% of Megachilidae depend on some type of non-floral, biotic resource (*Requier & Leonhardt, 2020*). Climate change is likely to only exacerbate the local levels of tree die-off in forested habitat, and in some instances, severe drought events can lead to over 90% tree die-off (*Breshears et al., 2005*). Additionally, an increase in wildfire frequency can lead to complete change in forest stands (*Flannigan, Stocks & Wotton, 2000*). It is unknown how these changes in forest structure will lead to subsequent changes in pollinator communities, especially within Megachilidae. Although little is known about climate change and soil composition, it is unlikely that ground nesting bees will have the same dramatic loss of nesting habitat under future climate scenarios (*Allen, Singh & Dalal, 2011*; *Rinot et al., 2019*). This uncertainty furthers

the need to quantify the amount of nesting resources needed for Megachilidae within local communities.

Understanding what factors influence the distribution and biogeography of this taxa is important for future conservation efforts. With megachilids in particular, it may not only be the direct effect of warming temperature that causes shifts in species distributions. These species have strong ties to their host plants which could cause phenological mismatch in the future if the bees and plants are not tracking climate in the same manner (*Bartomeus et al., 2018*; *Biesmeijer et al., 2006*). Future conservation efforts for this family should consider altogether the direct effects of warming temperature, availability of floral resources, and availability of nesting resources.

### Funding
This research was supported by the U.S. Department of Agriculture, Agricultural Research Service. The funders had no role in study design, data collection and analysis, decision to publish, or preparation of the manuscript.

### Grant Disclosures
The following grant information was disclosed by the authors:
U.S. Department of Agriculture, Agricultural Research Service.

### Competing Interests
The authors declare that they have no competing interests.

### Author Contributions
- Lindsie M. McCabe conceived and designed the experiments, performed the experiments, analyzed the data, prepared figures and/or tables, authored or reviewed drafts of the article, and approved the final draft.
- Paige Chesshire performed the experiments, authored or reviewed drafts of the article, and approved the final draft.
- Neil S. Cobb conceived and designed the experiments, authored or reviewed drafts of the article, and approved the final draft.

### Data Availability
The data is available at figshare: McCabe, Lindsie (2023). Megachilidae specimen records. figshare. Dataset. https://figshare.com/articles/dataset/Megachilidae_specimen_records/24084942/1.

### Supplemental Information
Supplemental information for this article can be found online at http://dx.doi.org/10.7717/peerj.16145#supplemental-information.

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
