# Peer review of "Forest habitats and plant communities strongly predicts Megachilidae bee biodiversity"

_PeerJ, doi:10.7717/peerj.16145_

## Round 0.1 · original submission · Major Revisions

Thanks for your submission to PeerJ. I really appreciate studies that draw from the growing amount of digitized biodiversity data, particularly in the context of better conservation and management. Rev2 also notes appreciation for this particular work.

Both reviewers provide some important criticisms of the study as currently presented, and those will require a fulsome and complete response – and likely a second round of review, hopefully by the same reviewers if willing – for further consideration of this MS.

In particular:

1. Rev1 notes that it is difficult to find the raw data. I did find a Dryad link in the associated material, but I'm not sure if Rev1 had access to that. In any case, please ensure that the full data are still available in your revision. It would also be good to at least rough-in a data availability statement in your M&M section (DOI).

2. Both Rev1 and Rev2 note that the authors need to make more complete use of the literature (taxonomy mentioned by Rev1 and nesting biology mentioned by Rev2). Further details in these areas would likely allow for some further analyses that would be useful for testing various hypotheses. Both reviewers make direct and indirect suggestions to this effect.

Both reviewers also provide a lot of other useful comments, including in a marked-up PDF from Rev1.

I believe that this paper has merit and likely utility for pollinator conservation, but there is still some substantial work to do to get it to a publishable state.

Please provide a line-numbered revision and a point-by-point rebuttal/response to the reviewers in the next iteration.

(I also noted the file listed as "Meg_Rich_MD_PLOS_nofigs.docx" which I assume means that this paper was initially submitted to PLOS. I didn't see a cover letter mentioning that. If that's the case, it might be useful to summarize what improvements have been made following the initial submission elsewhere.).

Reviewer 1 ·

Basic reporting

The authors cite a data set of >200,000 bee records. I did not see any way to access this. Raw data provided were just summarizing data.

They sometimes mix up things like the number of species / genera they are analysing and the actual number of species / genera that occur in the area. See note below on what that area actually is. It's not clear to me.

The authors have three primary questions. The first, relating to changes in megachilid diversity across 'North America' (read: USA). This has been well-established for decades. One only has to read Michener's classic 1978 paper on bee biogeography to know that the southwestern USA is a biodiversity hotspot. This also leads to a major design flaw, discussed in the next section.

The second and third questions, relates to what explains these patterns, related to climate, nesting, flowers, and wood plants. Again there is an issue here in study design that would need to be addressed.

There are some barplots that would probably be better as box plots. Their interpretation of their NMDS plot I can't quite fathom. They are reporting groups and connections that aren't at all evident to me. Table 1 is inconsistent, shows clear evidence of underlying biases in the raw data, and some obviously wrong data on 'average #'s of plant visited'. The authors will refer to 'known literature', but clearly haven't examined the most basic taxonomic literature for these groups to extract the relevant data, but are instead just skimming numbers from particular readily accessible databases.

Experimental design

It's unclear what data are being analysed at any time. They refer to downloading records from Canada, USA, and Mexico (btw this is not 'North America', which includes all the countries from Panama north, including Antillean islands). But at times it seems like this is just USA.

All bees are diverse in the southwestern USA, but few taxa have the same level of association with cavity-nesting as Megachilidae. So it is likely that at the coarse scale of analyses done here, one would find essentially the same results for Andrenidae, which have no cavity-nesting species at all. In order to establish that the there are specific influences true for cavity-nesting megachilid bees, the authors need to tease out the effects that are true for bees generally. I think you would need to figure out where Megachilidae are in greater diversity that one expects relative to other bees, and then establish that increased diversity is tied to woody plants. A real potential concern I have is whether the spatial scale allows for this. One can see on their coarse map, certain well known bee collection locations (e.g. Cochise County, Arizona). Based on my experience, this could be categorized as woody/shrubby habitat, due to the Larrea bushes in the desert and the heavy tree cover in the Chiricahua mountains, but the high diversity of bees (again, generally) one sees there is not strictly due to cavity-nesting locations that are available.

I have little doubt that high diversity of bees generally in the southwest is associated with climate and floral diversity, but there are historical biogeographic effects that need to be separated. There is immense topographical variability in the western USA that is not present in the east. Could this influence species diversity (of plants, bees, butterflies, ants, birds, fungi, etc.)?

There are some obvious sampling biases that can accrue with aggregated data sets, and I see little evidence that the authors have done anything to account for this. In their summary table, the show unexpectedly high counts for particular taxa (e.g. Ashmeadiella), which are likely due to revisionary studies or particular efforts to database a specific taxon. But this could unduly influence overall patterns. If there has been little effort to digitize bee data in the midwest, or to make those available on the data aggregators they skimmed, then it will a problem for down stream analyses.

Any attempt extract patterns from large data sets the authors have little control over runs the risk of 'garbage in, garbage out'. So there needs to be a more careful explanation of how issues with sampling bias have been accounted for. How reliable are the data? In my experience, looking at GBIF records of bees, only a small proportion meets basic quality standards (e.g. identified by a known expert on bees). There may be an increasingly high proportion stemming from iNaturalist on these sites, but that has its own biases, which are quite different from historical collection records.

Have the authors determined the specific nesting use of the 667 species they report? are there differences in responses between species that nest in the ground, on surfaces, in stems, in wood?

I think if one really wants to establish the effects of woody habitat on megachilid diversity, it's probably wrong to go to a coarse grained analysis. This seems like a question that would be better addressed at the local scale where these amounts of these resources could be assessed and compared more directly.

Validity of the findings

I think the authors are wrong to draw conclusions about the effects they are trying to establish. The southwest USA is a biodiversity hotspot for bees, regardless of their nesting preferences, so one can't draw conclusions from this pattern specific for cavity-nesting bees.

I think the general assumptions being made here about how suitable these data and analyses are for addressing the questions are flawed.

Data were not provided in full.

Additional comments

some more comments on the attached pdf.

Annotated reviews are not available for download in order to protect the identity of reviewers who chose to remain anonymous.

Reviewer 2 ·

Basic reporting

I think overall this is potentially a nice study, and is generally well-written, though with a few typos and issues in the formatting for the references, and had comments (some copied below). However, I have larger issues, which are discussed below, and feel that the minor comments are not as important until these other ones are addressed.

ln 54. rotundata, not rotunda (which is a non-native species)

ln 62. I am not sure that summarizing the nesting biology of the genera is as useful as accounting for the intrageneric variation, especially in the large genera (many subgenera of Megachile are ground nesters).

ln 67. widely accepted by whom? Please add references for such statements.

ln 183. Many species of Megachile in the USA are also ground nesters, including entire subgenera (e.g., Argyropile, Megachiloides, Xanthosarus).

ln 215, T'ai Roulston et al. did a paper a few years back (late 2000s) on importance of flowers versus nesting sites, that should be reviewed and cited.

Experimental design

My biggest issue with the manuscript is the broad categorization of nesting biology used at the genus level in Table 1, which was presumably what some of the analysis was based on. To me, there is so much variation in nesting biology within each genus that trends may not be as obvious, and the summaries defined are really not that helpful. For instance, does “branches and twigs” mean the bees nest on these (i.e., some anthidiines build resin-based nests as attachments) or in them? In addition, several subgenera of Megachile are exclusively ground nesters, and some species have flexibility and can use cavities in wood or the ground, some need decomposing wood. Cane et al. (2006) gave an excellent summary of the Osmia (e.g., under stones, snail shells, cavities) but this is not reflected in this study in Table 1. Why not use their study as a model and develop a summary table for all North American megachilids that summarizes what is known about each species, and then re-do your analysis based on more precise information…there is a lot of information to be gleaned from piles of literature. Although it would take a bit of time, I think the authors should really try to acquire as much species-level knowledge of nesting biology and recreate Table 1 as species specific, or perhaps even to subgenus at least. This would allow you to give more specific information – which makes me also ask, why does the phrase “species specific” apply for some genera, but not all of them? I think the approach used in table 1 is frankly, quite lazy, and I would not put any faith in the results of this study when nesting data of this general is used. I think the patterns of diversity in the study are well done, though I do not think the analysis is appropriate for so general summaries for nesting.

In addition, why not analyze the cleptoparasites separately, as their hot-spots should be correlated with hosts (and the nesting biology of the host).

Validity of the findings

Again, I question the findings when it comes to the use of such broad categories of nesting, and think the study would be really more valuable if the authors but more time in to gathering species specific data.

Additional comments

My first thought was to reject the MS, though I feel that it is potentially a vary nice study if better summaries of nesting biology were used. So I recommend major revisions.

---

## Round 0.2 · accepted · Accept

I would like to thank both reviewers and the co-authors of the manuscript for their work in improving this paper. Neither reviewer was able to re-review the paper, so I went through the revisions file and the paper and I am satisfied with the changes and responses by the co-authors to the reviewers' suggestions.

I do note, however, that while the co-authors provide a link to the underlying data in their response document, there is no DOI and/or URL listed in the paper or elsewhere. This will need to be rectified prior to publication, and I trust the authors can work with PeerJ staff to ensure readers have access to the data.